# Staged Sinus Floor Elevation Using Novel Low-Crystalline Carbonate Apatite Granules: Prospective Results after 3-Year Functional Loading

**DOI:** 10.3390/ma14195760

**Published:** 2021-10-02

**Authors:** Yoichiro Ogino, Yasunori Ayukawa, Noriko Tachikawa, Masahiro Shimogishi, Youji Miyamoto, Keiko Kudoh, Naoyuki Fukuda, Kunio Ishikawa, Kiyoshi Koyano

**Affiliations:** 1Section of Fixed Prosthodontics, Division of Oral Rehabilitation, Faculty of Dental Science, Kyushu University, Fukuoka 812-8582, Japan; ayukawa@dent.kyushu-u.ac.jp; 2Section of Implant and Rehabilitative Dentistry, Division of Oral Rehabilitation, Faculty of Dental Science, Kyushu University, Fukuoka 812-8582, Japan; 3Department of Oral Implantology and Regenerative Dental Medicine, Tokyo Medical and Dental University, Tokyo 113-8510, Japan; ntachikawa.impl@tmd.ac.jp (N.T.); shimogishi.irm@tmd.ac.jp (M.S.); 4Department of Oral Surgery, Institute of Biomedical Sciences, Tokushima University Graduate School, Tokushima 770-8504, Japan; miyamoto@tokushima-u.ac.jp (Y.M.); kkudoh@tokushima-u.ac.jp (K.K.); naoyukifukuda@tokushima-u.ac.jp (N.F.); 5Department of Biomaterials, Faculty of Dental Sciences, Kyushu University, Fukuoka 812-8582, Japan; ishikawa@dent.kyushu-u.ac.jp; 6Division of Advanced Dental Devices and Therapeutics, Faculty of Dental Science, Kyushu University, Fukuoka 812-8582, Japan; koyano@dent.kyushu-u.ac.jp

**Keywords:** low-crystalline carbonate apatite (CO_3_Ap), sinus floor elevation, functioning implants

## Abstract

The aim of this study was to evaluate clinical outcomes of staged sinus floor elevation (SFE) using novel low-crystalline carbonate apatite (CO_3_Ap) granules. Patients who needed SFE for implant placement were recruited into this clinical trial. A staged procedure (lateral window technique using CO_3_Ap granules, followed by implant placement after 7 ± 2 months) was employed in 13 patients. Bone-height increase and insertion torque values (ITVs) were assessed along with histological evaluation. The survival and success rates of 3-year functioning implants were also evaluated. Mean of bone-height increase after SFE using CO_3_Ap granules was 7.2 ± 2.5 mm and this increase allowed implant placement in all cases (17 implants). Mean of ITV was 25.1 ± 13.2 Ncm and primary stability was achieved successfully in all cases. Histological analyses revealed mature new bone formation (36.8 ± 17.3%) and residual CO_3_Ap granules (16.2 ± 10.1%) in the compartment after SFE. The survival and success rates after 3-year functional loading were 100% and no complications were found. These results clearly indicate the clinical usefulness of CO_3_Ap granules for SFE.

## 1. Introduction

Sinus floor elevation (SFE) has been reported to be a predictable treatment modality for implant placement in atrophic maxilla [1,2]. The SFE procedure includes two main techniques: the transcrestal (or transalveolar) technique and the lateral window technique [3]. The transcrestal technique using an osteotome approach is generally a less invasive procedure, has easier surgical technique, fewer complications and shorter surgical time [3,4]. On the other hand, the lateral window technique requires more invasive intervention and more skilled techniques [3,4,5,6]. However, this technique can be performed under direct vision and can increase more vertical space compared to the transcrestal technique [3,4]. Although the proper selection of surgical techniques is required depending on the anatomical situation, especially the residual bone height (threshold height: 4–6 mm) [4,5,6], both techniques were clinically reliable and predictable [7,8,9]. 

In SFE using the lateral window technique, various graft materials have been used and evaluated. Autogenous bone has been considered as the gold standard because of its osteoinductive and osteoconductive properties [10]. However, the disadvantages of autogenous bone are limited availability, postoperative morbidity and resorption [11,12,13]. As alternatives to autogenous bone, bone substitutes including allogenic bone, xenograft and alloplastic graft materials have been widely used and investigated [14,15,16,17,18]. In particular, alloplastic graft materials such as beta-tricalcium phosphate (β-TCP) and hydroxyapatite (HAp) have been used for SFE due to their potential capacities such as unlimited availability, ease of use and being less invasive. However, it should be noted that no artificial graft materials were approved in Japan for their clinical use in load-bearing area including implant-related bone reconstruction, due to their poor osteoconductivity when compared to autograft.

Since autograft is the gold standard, attempts to imitate bone seem reasonable to fabricate alloplastic graft materials with greater function. One of the key differences between bone and current alloplastic graft materials is the composition. While the composition of bone is carbonate apatite (CO_3_Ap), which contains approximately 6–9 mass% carbonate [19,20], CO_3_Ap decomposes during sintering procedures due to the presence of carbonate in apatite crystal. In contrast, HAp and β-TCP can be sintered and show osteoconductivity, leading their use as alloplastic graft materials. However, we succeeded in fabricating low-crystalline CO_3_Ap granules in an aqueous solution through a dissolution–precipitation reaction using a precursor such as calcium carbonate [21,22]. Histological analysis and cell study revealed higher osteoconductivity of CO_3_Ap when compared with other alloplastic graft materials [23,24,25]. Based on simulated clinical trials using beagle dogs, the first clinical trials of CO_3_Ap granules were performed in three university hospitals for SFE [26,27]. 

The aim of this study was to evaluate the follow-up results of staged SFE using novel CO_3_Ap granules with respect to bone height increase, insertion torque values (ITVs), replacement of CO_3_Ap to a new bone, and survival and success rates of the implants after functional loading. The null hypothesis was that staged SFE using CO_3_Ap granules did not have clinical availability for implant placement and occlusal loading.

## 2. Materials and Methods

### 2.1. Research Design and Ethical Approval

At first, this study was planned and conducted as a multicenter, single-arm, clinical trial in three university dental hospitals (Kyushu University Hospital, Tokyo Medical and Dental University Hospital, and Tokushima University Hospital). This clinical trial was conducted from January, 2015 to May, 2017 in accordance with the Declaration of Helsinki. The institutional review board approved this study as a clinical trial (clinical trial No. GCAP–01, #2014904). This clinical study was also registered as JPRN–UMIN000019281 in the University Hospital Medical Information Network in Japan as a clinical trial and the International Clinical Trials Registry Platform Search Portal of the World Health Organization. After the completion of this clinical trial, a subsequent study was planned and conducted as a prospective observational study. This observational study was approved by the institutional review board (#29–299, D2017-029 and 2927) and was conducted according to Strengthening the reporting of observational studies in epidemiology (STROBE). 

Prior to the conduction of this clinical trial, Pharmaceuticals and Medical Devices Agency (PMDA) in Tokyo, Japan advised us that at least 20 subjects would be necessary to assess the effect of CO_3_Ap granules on bone formation and the safety of CO_3_AP clinically. This clinical trial was planned to enroll at least 20 patients as subjects following PMDA’s suggestions.

The patients who needed SFE for implant placement could be candidates for this clinical trial. The candidates who met the criteria were enrolled as the subjects (Table 1). 

### 2.2. Surgical Procedure for Sinus Floor Elevation (SFE) and Implant Placement

In this study, the patients assigned to staged SFE (SFE using CO_3_Ap granules via lateral window technique followed by implant placement after healing of 7 ± 2 months) were evaluated and were followed up. 

The patients had computed tomography (CT) examinations with diagnostic stents prior to SFE (pre-SFE CT image). SFE with the graft of CO_3_Ap granules (GC Cytrans Granules, GC, Tokyo, Japan) was conducted under local infiltration anesthesia. The particle size of CO_3_Ap granules was in the range of 600–1000 μm. Briefly, the lateral wall of the sinus was exposed after the reflection of a full-thickness mucoperiosteum flap and a bony window was created. The Schneiderian membrane was separated carefully using sinus membrane elevators and the compartment was created by the sinus membrane elevation procedure. This compartment was filled with CO_3_Ap granules. The study protocol did not allow to use membrane to cover the lateral window. The first surgery was completed with the sutures (Figure 1).

After a healing period of 7 ± 2 months [13,15,16], a second CT examination with the same diagnostic stent was conducted to measure the bone height and to plan the implant placement (preimplantation CT image). Prior to implant placement, implant sites were prepared with a trephine bur (2.1 mm diameter) to obtain bone specimens for histological analysis. Implant placements were performed as manufacturers’ instructions. 

### 2.3. Implant Restorative Procedure

After 8 ± 2 months of healing period for osseointegration, a second surgery for uncovering the dental implant and connecting the abutment to the implant body was performed, and provisional restorations and definitive prostheses were fabricated as per the manufacturers’ instructions. All of the patients were asked to attend prospective observational studies, including regular implant maintenance programs, after informed consent.

### 2.4. Analyses of Outcomes

#### 2.4.1. SFE Evaluation in a Clinical Trial

This study was conducted as a first clinical trial, followed by a prospective observational study. The primary outcome of the clinical trial was the evaluation of ITVs (≥10 Ncm) as biomechanical characteristics of newly formed bone. The secondary outcome was the result of bone biopsy. The newly formed bone was assessed histologically. The composition of the newly formed bone (mature bone, osteoid and residual CO_3_Ap granules) was also evaluated with bone specimen after Villanueva–Goldner staining by well-experienced pathologists. Image J software (1.52u, 2020, National Institutes of Health, Bethesda, MD, USA) was used for histomorphometric analysis. In addition, bone height increase was calculated using pre-SFE and preimplantation CT images (the difference in bone height between two images) by a well-experienced radiologist using the OsiriX medical imaging software program (version 7.5, Open-Source, OsiriX Medical Imaging Software, http://www.osirix-viewer.com, accessed on 22 November, 19 and 26 December, 2016). The images (frontal views) used for analyses were selected based on the position of the diagnostic stents to match the images. Both analyses were performed in a blinded manner.

#### 2.4.2. Clinical and Radiographic Evaluations after 3-Year Functional Loading

Clinical and radiographic evaluations (panoramic radiography) were conducted at implant placement, at the delivery of final prosthesis and 3 years after the delivery. Implant survival rates and success rates based on criteria according to the previous studies [9,28,29] were assessed as a 3-year evaluation. The success criteria were as follows:No detectable mobility on clinical examination;No pain or other subjective sensation from the implant;No recurrent peri-implant infection or sign of peri-implantitis;No continuous radiolucency in the peri-implant bone.

A time-sequence chart of the procedures was shown (Figure 2). 

## 3. Results

### 3.1. Patients 

In total, 22 patients enrolled and had SFE using CO_3_Ap granules. Eight of 22 patients had SFE for simultaneous implant placement and 14 patients were assigned to the staged SFE procedure. One patient who had staged SFE was excluded from the analysis because the complex of CO_3_Ap granules and autogenous bone was used as graft materials for SFE. As a result, 13 patients (four males and nine females, mean age: 61.0 ± 11.7) and 17 implants were included in this study. The detailed data are described in Table 2. 

### 3.2. The Results of Surgical Procedures

During SFE, no perforation of the sinus membranes was detected. The CT images (frontal views) of pre-SFE and pre-implantation are shown in Figure 3. Table 2 also presents SFE sites and vertical bone heights measured using pre-SFE and pre-implantation CT images. The mean increase of bone height after SFE with CO_3_Ap graft was 7.1 ± 2.4 mm (mean ± standard deviation (SD)). They revealed that the CO_3_Ap graft could maintain the vertical height and the space between the elevated sinus membrane and sinus floor. The amounts of CO_3_Ap granules used for SFE are also presented in Table 2.

The implants were placed uneventfully in all cases. Detailed information regarding implants and ITVs, the primary outcome of this clinical trial, are described in Table 3. ITVs were not recorded in one patient with three implants. The mean ITV was 25.1 ± 13.2 Ncm (mean ± SD). All implants could acquire primary stability. 

A histological specimen (Villanueva–Goldner staining) is shown in Figure 4. In this staining, mature bone was stained in green, and osteoid was stained in red. New bone formation and residual CO_3_Ap granules were observed. Residual CO_3_Ap granules were surrounded by newly formed bone and osteoid was observed around new bone and CO_3_Ap granules. The specimens were used to calculate the ratio of bone area to total area. The composition of mature bone, osteoid and residual CO_3_Ap granules was calculated for histomorphometrical examinations. The mean ratios of mature bone, osteoid and residual CO_3_Ap granules were 37.6 ± 16.9, 2.4 ± 1.3 and 15.4 ± 10.1% (mean ± SD), respectively (Table 2). 

### 3.3. Clinical Outcomes

After implant placement, all implant-related surgeries (second operations: connection of healing abutments) and prosthetic procedures were completed successfully. Functional loading with definitive prostheses was possible in all cases. All implants in 12 patients have functioned for 3 years uneventfully. Unfortunately, definitive prosthesis in one patient did not achieve 3-year function, although the implant was loaded for more than 3 years with a provisional restoration. Implant survival rate was 100%. Implant success rate after 3-year functional loading was assessed with clinical findings and panoramic radiographic images based on the success criteria (Figure 5). All implants were stable and showed no mobility. No patients complained of pain or peri-implant soft tissue inflammation. In addition, panoramic radiography analyses showed no continuous radiolucency in the peri-implant bone. These findings revealed that implant success rate was also 100%. 

## 4. Discussion

Previous studies showed that SFE using the lateral window technique could achieve favorable clinical outcomes [1,2,7,9]. The lateral window technique was recommended in cases with less residual bone height (≤5–6 mm) [4,9], and delayed implant placement was recommended in the cases with residual bone height <4 mm [9,30,31]. In these cases, the lateral window technique using a graft material would be useful to maintain the space between the sinus membrane and floor, although a previous report suggested the effectiveness of SFE without any graft materials in space maintenance [32]. The most important factor in implant placement with SFE is to acquire the primary stability. To stabilize implants, sufficient bone height or volume, including pre-existing bone and newly formed bone, and mineralization, would be required. Previous studies revealed that SFE applying the lateral window technique with a graft material was a safe and predictable surgical procedure, especially in severe (less pre-existing bone height) cases [32,33,34]. This first clinical trial was conducted to evaluate the clinical availability of CO_3_Ap, which is a component of natural bone, as a graft material in SFE [30,31], and this study aimed to report the result after 3-year functional loading. The findings of this subsequent study have shown the clinical availability of CO_3_Ap granules for SFE and implant treatment.

This clinical trial was planned to measure the vertical height at implant site (indicated by a diagnostic stent in CT image) to assess the result of SFE. The mean value of vertical bone height prior to SFE was 3.5 ± 1.3 mm. All sites were less than 6 mm and 10 sites were less than 4 mm, which were recommended to receive delayed implant placement [9,30,31]. CT images revealed that CO_3_Ap granules worked effectively as a graft material in SFE and as materials for space maintenance and new bone formation (7.1 ± 2.4 mm). However, future CT evaluation of three-dimensional mineralization after SFE (vertical and horizontal dimension) would be favorable. The histological analyses showed new bone formation and bone mineralization. The formation of osteoid around residual CO_3_Ap granules, which was similar to newly formed bone, suggested the capacity for bone formation by CO_3_Ap granules. ITVs revealed the primary stability in all cases. In addition, all implants and prostheses functioned uneventfully after 3-year functional loading (survival and success rates: 100%) and these results suggested the availability of CO_3_Ap granules as a graft material in SFE and delayed implant placement.

Previous studies have reported the results of SFE with various graft materials. They suggested that the type of graft materials were not assumed to be associated with clinical outcomes [11,14]. We would like to focus attention on the characteristics of CO_3_Ap granules. The apatite in human bone is CO_3_Ap, not hydroxyapatite (HAp) [19,20]. In comparison with HAp in vitro, CO_3_Ap is resorbed by osteoclasts [35] and enhances osteoblastic activities [25]. CO_3_Ap fabricated by the dissolution–precipitation reaction showed more bone replacement compared to Bio–Oss [23]. The evaluation of bone formation using three commercially available bone substitutes with different composition (CO_3_Ap, HAp and β-tricalcium phosphate) revealed that CO_3_Ap demonstrated the highest level of new bone formation and suggested that similar composition of bone could contribute to bone formation [36]. In addition, SFE using CO_3_Ap granules in simultaneous and delayed implant placements has been shown to achieve bone-height increase and successful osseointegration [26,27]. These studies revealed the availability of CO_3_Ap granules as a graft material in SFE and implant placement experimentally and clinically. This study could demonstrate the clinical effectiveness of CO_3_Ap granules in SFE after 3-year functional loading, which is necessary for implant restoration. We believe that this study encourages the clinical application of CO_3_Ap granules in SFE and implant placement.

Prior to conclusion of this study, it is required to mention its limitations. Firstly, the clinical assessment was limited to small sample size. This study was conducted as an observational study following a first clinical trial [27]. As mentioned above, we needed to enroll 22 patients to conduct a clinical trial without any control groups, and 14 of them were assigned to staged SFE. Although studies to evaluate the availability of CO_3_Ap granules as a graft material will be reported in the near future, this study could provide important data on the clinical availability of CO_3_Ap granules after functional loading. Secondly, detailed analyses such as the change in peri-implant bone level were not conducted. The change in peri-implant bone level has been known to be affected by multiple factors [37,38,39,40]. The design of this study was SFE, in which bone resorption might occur at coronal (preexisting bone) and apical (augmented bone) sites. The aim of this study was to evaluate the clinical availability of CO_3_Ap granules and the prognosis of the implants (the survival and success rates of implants in newly formed bone by CO_3_Ap granules) was reported. These success criteria, including radiographic evaluations, were widely adopted as the evaluation of implant prognosis after functional loading [9,28,29], although other success criteria have been widely used [41]. This study did not evaluate peri-implant bone level using standardized X-ray images and it was difficult to evaluate the change of bone level precisely. The previous studies also reported survival and success rates of implants placed after SFE [1,2,4,11,13,15]. No remarkable bone resorption around implants was observed radiographically and all implants were defined as successful implants clinically. Whereas this study suggested the clinical availability of CO_3_Ap granules used in SFE, future studies will be required to evaluate the effect of CO_3_Ap granules on new bone formation and temporal changes of newly formed bone around implants, not only in SFE but also in bone augmentation around the implants. Finally, temporal change of CO_3_Ap granules could not be analyzed in detail. The histological analyses at implant placement showed that the new bone formation and the residual CO_3_Ap granules depended on each subject. The statistical correlation analysis among the ratio of new bone formation, residual CO_3_Ap granules and numerical data within the small sample size identified no significant correlation (data not shown), which suggested that contributing factors for bone formation in SFE using CO_3_Ap granules still remain unknown. In addition, detailed bone formation after implant placement could not be assessed. A well-designed future study should be considered to estimate the temporal changes of bone quality and quantity formed by CO_3_Ap granules.

## 5. Conclusions

This study revealed that CO_3_Ap granules formed new bone after SFE, and the newly formed bone supported implants successfully after 3-year functional loading, suggesting the clinical availability of CO_3_Ap granules in implant rehabilitation. Further studies are required to validate the factors related to new bone formation clinically, and to compare the bone formation between CO_3_Ap granules and other graft materials.

## Figures and Tables

**Figure 1 materials-14-05760-f001:**
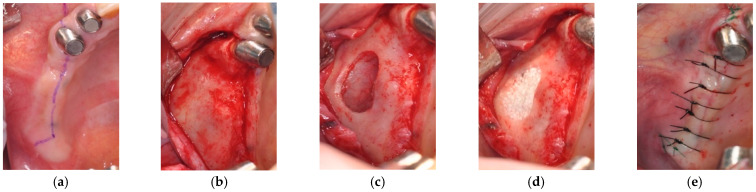
Sinus floor elevation (SFE) with lateral window technique: (**a**) Preoperative intraoral image; (**b**) Exposure of lateral sinus wall; (**c**) Sinus membrane elevation procedure from lateral sinus wall; (**d**) Graft of CO_3_Ap granules; (**e**) Completion of SFE with sutures.

**Figure 2 materials-14-05760-f002:**
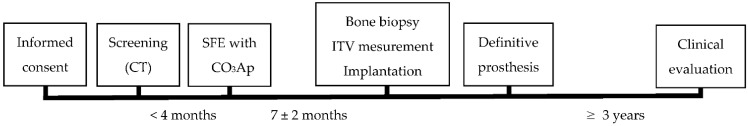
A time-sequence chart of the procedures. CT: computed tomography; SFE: sinus floor elevation; CO_3_Ap: carbonate apatite; ITV: insertion torque value.

**Figure 3 materials-14-05760-f003:**
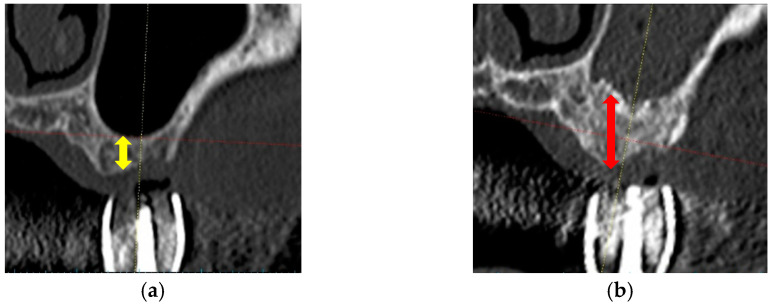
CT images of pre-SFE and pre-implantation. (**a**) Pre-SFE (frontal view), the yellow allow shows pre-vertical bone height; (**b**) Pre-implantation (frontal view), the red allow shows post-vertical bone height.

**Figure 4 materials-14-05760-f004:**
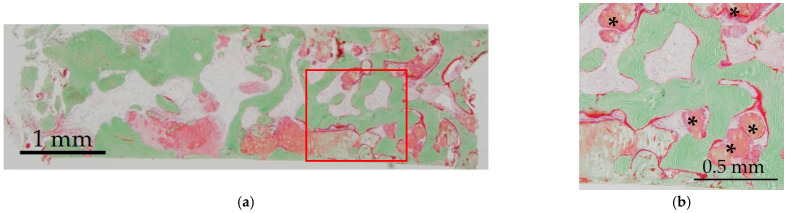
Histological analysis of newly formed bone after SFE (Villanueva–Goldner staining). Mature bone is stained green and osteoid is stained red: (**a**) Low magnification; (**b**) Higher magnification of red square in (**a**). *: residual CO_3_Ap granules.

**Figure 5 materials-14-05760-f005:**
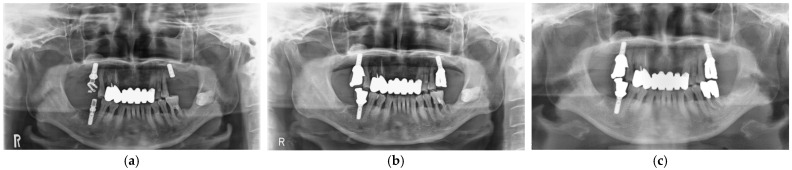
Panoramic radiographic images. SFE and implant at maxillary left molar site. (**a**) At implant placement; (**b**) At the delivery of final restoration; (**c**) At 3-year follow-up.

**Table 1 materials-14-05760-t001:** Inclusion and exclusion criteria.

**Inclusion Criteria**
Partial edentulous maxilla (premolars and/or molars) Residual alveolar bone height of less than 5 mm from the original sinus floor to the crest of the alveolar bone in computed tomography images Age: between 20 and 80 years old
**Exclusion Criteria**
Presence or history of malignant tumor-related therapy (radiotherapy and/or chemotherapy) Uncontrolled diabetes mellitus History of bone metabolism disease (ex: osteoporosis) and related medication (ex: bisphosphonate) Immunodeficiency, infectious disease, or connective tissue disease Immunosuppressant medication (excluding local administration) Severe kidney, liver, blood, bone metabolism or circulatory system disorder that prevents dental surgery Pregnancy, possibly pregnancy, breastfeeding, or considering pregnancy Alcohol or drug abuse Psychological or psychiatric disorders Artificial dialysis Maxillary sinus pathologies Oral infections or uncontrolled periodontal disease Participation in another clinical trial program within 3 months prior to study entry

**Table 2 materials-14-05760-t002:** Patient’s profiles and the data of SFE and bone biopsy.

Age and Gender	Site	Pre-SFE Bone Height (mm)	CO_3_AP Amount (cm^3^)	Pre-Implant Placement Bone Height (mm)	Histomorphometrical Analyses (%)		
					Mature Bone	Osteoid	CO_3_AP
71, Female	16	5.0	1.2	8.8	10.7	0.7	23.1
	17	3.8		14.5			
64, Male	26	5.0	1.5	8.1	58.2	1.6	14.0
69, Female	25	4.4	0.8	13.3	24.6	0.6	1.3
43, Female	26	5.0	0.7	13.3	23.1	1.5	34.8
64, Female	25	2.6	1.9	13.0	55.7	1.9	24.8
77, Female	27	3.6	1.0	7.5	47.5	1.4	9.2
56, Male	16	3.3	1.5	11.2	48.5	3.0	13.6
67, Female	25	5.0	1.4	9.0	35.7	2.4	4.8
	26	2.0		10.3			
71, Male	26	4.0	1.4	12.0	30.1	0.6	7.5
60, Female	25	4.1	1.8	9.3	49.8	0.6	18.3
37, Female	15	3.6	0.4	12.7	24.6	3.6	12.7
64, Male	26	2.0	0.8	8.4	59.1	5.2	15.0
50, Female	14	3.0	2.5	8.9	11.1	3.4	31.6
	15	1.4		9.8			
	16	1.0		11.1			
Mean ± SD		3.5 ± 1.3	1.3 ± 0.6	10.7 ± 2.1	36.8 ± 17.3	2.0 ± 1.4	16.2 ± 10.1

SFE: sinus floor elevation; CO_3_Ap: carbonate apatite; SD: standard deviation.

**Table 3 materials-14-05760-t003:** Data on implant placement.

Age and Gender	Site	ITV (Ncm)	Implant		
			Company	Diameter (mm)	Length (mm)
64, Male	26	30	SM	4.8	8.0
71, Female	16	14	SM	4.1	8.0
	17	15	SM	4.1	10.0
69, Female	25	15	NB	4.0	10.0
43, Female	26	22	DP	4.5	9.0
64, Female	25	14	NB	3.8	10.0
77, Female	27	20	GC	4.4	8.0
56, Male	16	49	SM	4.1	8.0
67, Female	25	12	NB	4.3	8.5
	26	13	NB	4.3	11.5
71, Male	26	50	GC	4.4	10.0
60, Female	25	36	SM	4.1	8.0
37, Female	15	37	GC	3.8	8.0
64, Male	26	25	SM	4.8	8.0
50, Female	14	NR	GC	3.8	10.0
	15	NR	GC	3.8	10.0
	16	NR	GC	3.8	8.0

ITV: insertion torque value; NR: not recorded; SM: Straumann AG, Basel, Switzerland; NB: Nobel Biocare AB, Göteborg, Sweden; DP: Dentsply Implants, Mölndal, Sweden; GC: GC Corporation, Tokyo, Japan.

## Data Availability

All the data is available within the manuscript.

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
