# Peer review of "Staged Sinus Floor Elevation Using Novel Low-Crystalline Carbonate Apatite Granules: Prospective Results after 3-Year Functional Loading"

_materials, 2021, doi:10.3390/ma14195760_

Round 1

Reviewer 1 Report

This case series study evaluates bone formation from CO3AP. To do this, it uses an SFE design and a conversion time of 7 +/- 2 months before the biopsy. At the same time, it evaluates the survival / success of the implants in the SFE procedure.

Recommendations:

The study must clearly state that it is a case series.
This study would benefit from introducing a time sequence chart of the procedures performed. (see attached figure)

Author Response

Reviewer 1

This case series study evaluates bone formation from CO3AP. To do this, it uses an SFE design and a conversion time of 7 +/- 2 months before the biopsy. At the same time, it evaluates the survival / success of the implants in the SFE procedure.

Recommendations:

The study must clearly state that it is a case series.                                         

>We appreciate your comments. As you pointed out, this is exactly a case series. However, this study was conducted as a clinical trial to evaluate the effect of CO3AP on bone formation for SFE. We clearly mentioned in the text. The primary outcome of this clinical trial was insertion torque value and secondary outcome was bone biopsy. In addition, all patients had CT evaluation before SFE and before implantation, bone height was also evaluated. (These were added in the text as other reviewer suggested.) After that, this study was conducted as a prospective observational study. As other reviewer recommended, the title was changed into ““Staged Sinus Floor Elevation Using Novel Low–Crystalline Carbonate Apatite Granules: Prospective results after 3-years of functional loading” and we believe that the design of this study was clearly indicated. I hope you understand our intension. And we need to tell you one thing. We’ve checked our data again and we’re afraid that one patient was excluded because her definitive prosthesis did not reach 3-year function, although the implant worked more than 3 years with provisional restoration and did not show any problems. Please understand this situation.

This study would benefit from introducing a time sequence chart of the procedures performed. (see attached figure)

> We can agree your comment. The chart was added as Figure and the following figures were renumbered.                                                                                                    

Reviewer 2 Report

The study describes the use of low–crystalline Carbonate Apatite Granules for external sinus-floor elevation. 13 patients were treated and received subsequently 17 implants after 8 months of graft healing. The follow-up period was 3 years.

There are some points of the manuscript that need further clarification:

The number of patient seems to be relatively small compared to current studies in this field. Why were more patients not included? Was a sample size calculation performed?

Which parameter was the primary outcome criterion and which parameters were secondary outcome criteria (Insertion torque values? Histology? Histomorphometry? Vertical bone gain? Volume gain?). The primary outcome also required for sample size calculation.

Why was not a control group included? Especially when the gold standard in Japan is autogeneous bone and there are no other bone graft substitutes available as an alternative?

How many observers performed the outcome analysis? How was calibration performed (Inter-observer reliability)? Were the observers involved in the treatment of the patients?

The authors should report their data according to STROBE guidelines.  Please check: https://www.strobe-statement.org/index.php?id=available-checklists and include all points in the manuscript. Furthermore include a completed checklist as supplemental material.

Where are the data on the radiographic volume increase measured in the CTs? Please add this data, especially considering that the 3D data is available. It is not sufficient to add only the vertical bone bone gain.

 Was a membrane used to cover the lateral window after sinus augmentation? If yes please complete the surgical description.

The duration of 8 months bone healing seems very long. What was the reason for this time period? Is this the time the authors let autogenous bone heal as well? There are studies with healing periods of 4 months with osteconductive alloplastic materials. Please add this to the discussion.

The authors use success criteria for implants that do not include values for marginal bone stability. The authors should use the well-established criteria of Albrektsson. They are the standard for most studies.

It should be added in the introduction that the selection of lateral approach or transcrestal approach is dependent upon on the remaining vertical bone height. Systematic reviews describe less favorable outcome with the transcrestal approach when the residual bone height is less than 4mm. This fact is correctly addressed by the authors in the discussion section, but should also be mentioned at the introduction.

The sentence in lines 79-81 is redundant and should be deleted. This fact was already mentioned in the introduction (lines 55-56).

The magnification is missing in the caption of figure 3. Low magnification and high magnification is not sufficient as description.

The title should be altered: The current version (“Clinical Outcomes of Staged Sinus Floor Elevation Using Novel Low–Crystalline Carbonate Apatite Granules and 3–year Functioning Implants”) is not accurate. The authors not only report clinical outcomes but also histomorphometric and radiographic outcomes.

The title could be named:

“Staged Sinus Floor Elevation Using Novel Low–Crystalline Carbonate Apatite Granules: Prospective results after 3-years of functional loading”

Abstract:

In the abstract the part of the sentence “and 3–year functioning implants” should be deleted (line 22). It was not the aim of this study.

Author Response

Reviewer 2

The study describes the use of low–crystalline Carbonate Apatite Granules for external sinus-floor elevation. 13 patients were treated and received subsequently 17 implants after 8 months of graft healing. The follow-up period was 3 years.

There are some points of the manuscript that need further clarification:

The number of patient seems to be relatively small compared to current studies in this field. Why were more patients not included? Was a sample size calculation performed?                   

>Thank you for your comment. As we mentioned in this study, this study was conducted as a clinical trial to evaluate the effect of CO3AP on bone formation for SFE and the safety of CO3AP clinically. Pharmaceuticals and Medical Devices Agency (PMDA) in Japan suggested at least 20 cases would be necessary to evaluate for this purpose. We had 22 subjects and 8 of them were treated using 1-staged SFE. Fourteen subjects had 2-staged SFE and one of them had SFE with CO3AP combined with autogeneous bone. This subject was excluded from this analysis to evaluate equally. These detailed information was described in 2. Materials and Methods, 2.1. Research design and ethical approval, and 3. Results, 3.1. Patients. In addition, we’ve already described some comments in 4th paragaraph, Discussion. The further detailed descriptions were added in Discussion.

Which parameter was the primary outcome criterion and which parameters were secondary outcome criteria (Insertion torque values? Histology? Histomorphometry? Vertical bone gain? Volume gain?). The primary outcome also required for sample size calculation.                

> The primary outcome of this clinical trial was insertion torque value, which was more than 10 Ncm. The value of 10 Ncm was enough for primary stability based on our experience. This was shown by our article written in Japanese. The secondary outcome was bone biopsy. In addition, all patients had CT evaluation before SFE and before implantation, and the protocol of this clinical trial also included bone height evaluation. Regarding the sample size, as we mentioned as the response to other reviewers, this study was planned as a clinical trial at first and PMDA advised us we needed at least 20 subjects for the purpose of our trial. Unfortunately, we understood that we needed sample size calculation and the sample size was not enough for the study, but we hope that you can understand that this prospective observational study was conducted following a first clinical trial. These comments were added in 2. Materials and Methods, 2.1. Research design and ethical approval.

Why was not a control group included? Especially when the gold standard in Japan is autogeneous bone and there are no other bone graft substitutes available as an alternative?  

>As we’ve mentioned in the text, no other bone graft substitutes were available officially when we started this clinical study (Ministry of Health, Labour and Welfare in Japan did not approve the graft materials for SFE officially). As you suggested, autogeneous bone was the gold standard in Japan as graft materials. However, graft of autogeneous bone is very invasive as a clinical trial because we needed donor site and to conduct a clinical trial, more 20 patients would be required to compare between 2 groups. To enroll and follow a total of 40 patients, more time and expense would be required and we gave up to set up control group. These comments were added in 4th paragraph, Discussion.

How many observers performed the outcome analysis? How was calibration performed (Inter-observer reliability)? Were the observers involved in the treatment of the patients?           

> We offered the radiologist for bone height evaluation and the pathologist for bone biopsy. Of course, they were well-experienced ones and not involved in the treatment of the patients. These analyses were blinded analyses. Some comments were added in 2. Materials and Methods, 2.4. Analyses of outcomes, 2.4.1. SFE evaluation in a clinical trial.

The authors should report their data according to STROBE guidelines.  Please check: https://www.strobe-statement.org/index.php?id=available-checklists and include all points in the manuscript. Furthermore include a completed checklist as supplemental material.                   

>This comment is very valuable. We followed your comment and we added some comments in 2. Materials and Methods,2.1. Research design and ethical approval and a completed checklist as supplemental material (circle of the number was checked and x means “not correspond”).

Where are the data on the radiographic volume increase measured in the CTs? Please add this data, especially considering that the 3D data is available. It is not sufficient to add only the vertical bone gain.                                                                  

>We applied the study protocol which evaluated bone height at implant site (the stent could be indicated this site), but not included 3D evaluation, unfortunately. To place the implants, bone height evaluation was more important because the main purpose of SFE to change vertical dimension. This is not scientific opinion, but in many cases, implant placement can be possible when we had sufficient bone height. If horizontal augmentation was conducted (mesially or distally), 3D evaluation was great, but we may need more vertical augmentation or elevation for implant placement. I understand this does not sound scientifically. Some comments were added in 2nd paragraph, Discussion.

 Was a membrane used to cover the lateral window after sinus augmentation? If yes please complete the surgical description.                                                               

>This must be described in the text. Coverage of the lateral window with membrane was not allowed in this study. We added the description in 2. Materials and Methods, 2.2. Surgical procedure for SFE and implant placement.

The duration of 8 months bone healing seems very long. What was the reason for this time period? Is this the time the authors let autogenous bone heal as well? There are studies with healing periods of 4 months with osteconductive alloplastic materials. Please add this to the discussion.                                                                         

>At first, we made an error in Abstract and 2. Materials and Methods, 2.2. Surgical procedure for SFE and implant placement. The healing time after SFE was 7 ± 2 months and we revised. The healing period for SFE was decided according to the previous studies. The study protocols of Ref# 13,15 and 16 were from 6 to 9 months. They also evaluated the newly formed bone histomorphometrically. We followed these protocols. We added some comments in Discussion.

The authors use success criteria for implants that do not include values for marginal bone stability. The authors should use the well-established criteria of Albrektsson. They are the standard for most studies.                                                            

>We can agree your comment. To apply this criteria, we need to measure peri-implant bone level using standardized X-ray images. In this study, we did not perform this method. So we used other criteria, bur more clinical observation was adopted in this study. We hope that you understand our protocol. We added some comments in 4th paragraph, Discussion.

It should be added in the introduction that the selection of lateral approach or transcrestal approach is dependent upon on the remaining vertical bone height. Systematic reviews describe less favorable outcome with the transcrestal approach when the residual bone height is less than 4mm. This fact is correctly addressed by the authors in the discussion section, but should also be mentioned at the introduction.                                                     

>Thank you for your comment. We would describe this condition as the anatomical situation, but detailed value (threshold height: 4–6 mm) was added in Introduction.

The sentence in lines 79-81 is redundant and should be deleted. This fact was already mentioned in the introduction (lines 55-56).                                                >Thank you for your comment. We described here because we think that this was necessary not to set up control group. However, we added other reasons following your suggestions and we agree this comment.

The magnification is missing in the caption of figure 3. Low magnification and high magnification is not sufficient as description.                                                         

>We think and believe that the magnification was meaningless because the scale of the figure was changed in the text. That’s why we added scale bar in the figures. I believe you can understand our intention.

The title should be altered: The current version (“Clinical Outcomes of Staged Sinus Floor Elevation Using Novel Low–Crystalline Carbonate Apatite Granules and 3–year Functioning Implants”) is not accurate. The authors not only report clinical outcomes but also histomorphometric and radiographic outcomes.

The title could be named:

“Staged Sinus Floor Elevation Using Novel Low–Crystalline Carbonate Apatite Granules: Prospective results after 3-years of functional loading”                                     >We really appreciate your suggestion. We changed the title.

Abstract:

In the abstract the part of the sentence “and 3–year functioning implants” should be deleted (line 22). It was not the aim of this study.                                                 

>We deleted as you suggested.

Reviewer 3 Report

Congratulations on this excellent paper !

As you already mentioned some more studies should be undertaken with a higher number of patients. You ought to ask some statisticians on how many cases you need for a high power of your results.

Minor refers only to the few mistakes in the English language . In one sentence in the beginning the authors used „less“ where „fewer“ would have been the correct word like less milk but fewer numbers.   A native English speaker will be able to detect these remaining minor mistakes. 

Author Response

Reviewer 3

Congratulations on this excellent paper !

As you already mentioned some more studies should be undertaken with a higher number of patients. You ought to ask some statisticians on how many cases you need for a high power of your results.                                                                         

>Thank you for your comment. Other reviewers also commented. Actually, this study was composed of a clinical trial and a subsequent prospective observational study. When we planned a clinical trial, Pharmaceuticals and Medical Devices Agency (PMDA) in Japan suggested at least 20 cases would be necessary to evaluate for this purpose. We had 22 subjects and 8 of them were treated using 1-staged SFE. Fourteen subjects had 2-staged SFE and one of them had SFE with CO3AP combined with autogeneous bone. This subject was excluded from this analysis to evaluate equally. These detailed information was described in 2. Materials and Methods, 2.1. Research design and ethical approval, and 3. Results, 3.1. Patients. In addition, we’ve already described some comments in 4th paragaraph, Discussion. The further detailed descriptions were added in Discussion.

Minor refers only to the few mistakes in the English language . In one sentence in the beginning the authors used „less“ where „fewer“ would have been the correct word like less milk but fewer numbers. A native English speaker will be able to detect these remaining minor mistakes.                                                                            

>Thank you for your comment. We checked again and some revisions were done.

Reviewer 4 Report

This article evaluates the clinical outcomes of a new bone regeneration material (CO3Ap granules) used in maxillary sinus elevation using the lateral window technique. In general the article is well structured and well written.

Some notes:

The sample size should be mentioned in the abstract. It is necessary to reach the results to realize that the sample comprises 13 patients.

Were all surgeries performed by the same operator?

What program was used to compare pre- and post-op CBCT? Has the examiner been calibrated? What reference points were used for the measurements? In figure 2 the measurements apparently are not made in the same CBCT section.

Were any precautions taken to ensure the reproducibility of the imaging exams, particularly with regard to  patient positioning?

The use of panoramic radiographs in postoperative controls has limitations, being preferable to use periapical x-rays.

In addition to the insertion torque, it would have been interesting to carry out an RFA to assess the ISQ.

  In addition to the reduced sample size the lack of a control group is one of the major limitations of this study, however the authors present a valid justification for its absence. 

Author Response

Reviewer 4

This article evaluates the clinical outcomes of a new bone regeneration material (CO3Ap granules) used in maxillary sinus elevation using the lateral window technique. In general the article is well structured and well written.

Some notes:

The sample size should be mentioned in the abstract. It is necessary to reach the results to realize that the sample comprises 13 patients.                                            >Thank you for your comment. We described clearly.

Were all surgeries performed by the same operator?                                      >No. This was a multicenter study and several surgeons performed SFE.

What program was used to compare pre- and post-op CBCT? Has the examiner been calibrated? What reference points were used for the measurements? In figure 2 the measurements apparently are not made in the same CBCT section.                          >As other reviewer commented similar ones, we added some comments including software for evaluation. The measurement was performed by a well-experienced radiologist. Figure 2 was modified. The radiologist used only frontal view images to measure vertical height at implant sites based on the stent position. We deleted sagittal view to make simple. Please refer some comments in the text, 2. Materials and Methods

Were any precautions taken to ensure the reproducibility of the imaging exams, particularly with regard to patient positioning?                                                       

>Unfortunately, CT exposure was not standardized, However, all patients were exposed with stent for imaging and we used very close position as much as possible. I hope you can understand our effort. We added some comments regarding the stent usage in the text.

The use of panoramic radiographs in postoperative controls has limitations, being preferable to use periapical x-rays.                                                       >Completely you’re right. To measure peri-implant bone level, periapical x-rays must be favorable. However, we’d like to evaluate bone level change from apical and coronal sites of implant, meaning that panoramic radiographs were better for SFE evaluation. We added some comments as limitation of this study in Discussion.

In addition to the insertion torque, it would have been interesting to carry out an RFA to assess the ISQ.

  In addition to the reduced sample size the lack of a control group is one of the major limitations of this study, however the authors present a valid justification for its absence.            

>This was also good point. Actually, in Japan, most clinicians understand that RFA and ISQ have clinical availability. However, these concept is not well used in Pharmaceutical Affairs Law in Japan. This study was started as a clinical trial and we decided to use ITV. As you commented, we added some comments regarding sample size in the text. We hope you can understand our design.

Round 2

Reviewer 2 Report

The authors have clarified some aspects of the previous review round, but some points require further minor changes.

Success criteria nowadays include relevant factors such as marginal bone stability (e.g. Albrektsson et al. 1986) or soft-tissue parameters as probing depth or bleeding on probing (e.g. Karoussis et al. 2004). These are important factors that have an influence on further prognosis. The factors described and used by the authors are essentially implant survival. So if the authors do not have the information on marginal bone stability or soft tissue parameters I would not report the current findings as success rate. They can still report their findings, but I would refrain from describing a success rate of 100% in this study. Nevertheless, I am confused as to what the authors are recording at their follow-up appointments. If neither intraoral radiographs for marginal bone stability nor soft-tissue parameters are recorded -which are both important indicators for screening periimplantitis- what are the authors recording in their follow-up appointments?

The authors state that torque at implant insertion was the primary outcome parameter. If this is the case, the authors should describe these in results section first before proceeding to the secondary outcome parameter.

Since the CT data of the 13 included patients are available and the authors have already used the Osirix software they should also report on the volume gain achieved by the bone augmentation procedure and not only the vertical height. This would be particularly interesting since the authors recorded the exact amount of graft material used (table 2).

Furthermore, it is “Osirix” software not “Osiris” (line 139)

The newly inserted text is partly not very scientifically written (e.g. line 265-272).

Author Response

The authors have clarified some aspects of the previous review round, but some points require further minor changes.

 >Thank you for your comments. I understand that your intentions and we’d like to describe our opinion.

Success criteria nowadays include relevant factors such as marginal bone stability (e.g. Albrektsson et al. 1986) or soft-tissue parameters as probing depth or bleeding on probing (e.g. Karoussis et al. 2004). These are important factors that have an influence on further prognosis. The factors described and used by the authors are essentially implant survival. So if the authors do not have the information on marginal bone stability or soft tissue parameters I would not report the current findings as success rate. They can still report their findings, but I would refrain from describing a success rate of 100% in this study. Nevertheless, I am confused as to what the authors are recording at their follow-up appointments. If neither intraoral radiographs for marginal bone stability nor soft-tissue parameters are recorded -which are both important indicators for screening periimplantitis- what are the authors recording in their follow-up appointments?

>We can completely agree your comment. As we wrote, we’ve checked the panoramic X-ray to observe the results of SFE. As you mentioned, we observed marginal bone using panoramic X-ray and soft tissue condition. We adopted the success criteria as we wrote and the previous studies used these criteria to assess the success rates. We believe that these criteria include the observational points you indicated. Of course, some patients had intraoral radiographs, but not all. I hope you can understand our criteria.

The authors state that torque at implant insertion was the primary outcome parameter. If this is the case, the authors should describe these in results section first before proceeding to the secondary outcome parameter.

>Thank you for your comments. The results were shown following the time course. We decided that it was better to show the results using table in this manner. However, we added in Results that ITVs were the primary outcome in this clinical trial according to your advice. We hope you understand our thoughts.

Since the CT data of the 13 included patients are available and the authors have already used the Osirix software they should also report on the volume gain achieved by the bone augmentation procedure and not only the vertical height. This would be particularly interesting since the authors recorded the exact amount of graft material used (table 2).

Furthermore, it is “Osirix” software not “Osiris” (line 139)

 >Thank you for your comments. We completely understand your comments. As we wrote as last comments, we applied the study protocol which evaluated bone height at implant site (the stent could be indicated this site), but not included 3D evaluation, unfortunately. If we evaluate 3D evaluation, we need to apply or modify our study protocol in each university. It will take same time. Of course, we truly understand your comment, but I would like you to understand our situation. Osiris was changed into Osirix.

The newly inserted text is partly not very scientifically written (e.g. line 265-272).

>Thank you for your comment. We added to describe the reason why we did not have the control group. However, we deleted following your comment.